# Runoff and Sediment Yield Processes in a Tropical Eastern Indian River Basin: A Multiple Machine Learning Approach

Alireza Moghaddam Nia [1], Debasmita Misra [2,*], Mahsa Hasanpour Kashani [3], Mohsen Ghafari [4], Madhumita Sahoo [5], Marzieh Ghodsi [6], Mohammad Tahmoures [1], Somayeh Taheri [1] and Maryam Sadat Jaafarzadeh [1]

[1] Faculty of Natural Resources, University of Tehran, Karaj 3158777871, Iran; a.moghaddamnia@ut.ac.ir (A.M.N.); m.s.jaafarzadeh@ut.ac.ir (M.S.J.); tahmoures@ut.ac.ir (M.T.); stahery@ut.ac.ir (S.T.)

[2] Department of Civil, Geological and Environmental Engineering, College of Engineering and Mines, University of Alaska Fairbanks, P.O. Box 755800, Fairbanks, AK 99775, USA

[3] Department of Water Engineering, Faculty of Agriculture and Natural Resources, University of Mohaghegh Ardabili, Ardabil 676W+5CX, Iran; m.hkashani@uma.ac.ir

[4] Department of Range and Watershed Management, Faculty of Water and Soil, University of Zabol, Zabol 3585698613, Iran; m.ghafari@frw.ir

[5] Department of Mining and Geological Engineering, College of Engineering and Mines, University of Alaska Fairbanks, P.O. Box 755800, Fairbanks, AK 99775, USA; sahoomadhu1989@gmail.com

[6] Faculty of Geography, University of Tehran, Tehran 1417853933, Iran; mghodsi@ut.ac.ir

* Correspondence: dmisra@alaska.edu; Tel.: +1-907-474-5339

**Abstract:** Tropical Indian river basins are well-known for high and low discharges with high peaks of flood during the summer and the rest of the year, respectively. A high intensity of rainfall due to cyclonic and monsoon winds have caused the tropical Indian rivers to witness more runoff. These rivers are also known for carrying a significant amount of sediment load. The complex and non-linear nature of the sediment yield and runoff processes and the variability of these processes depend on precipitation patterns and river basin characteristics. There are a number of other elements that make it difficult to forecast with great precision. The present study attempts to model rainfall–runoff–sediment yield with the help of five machine learning (ML) algorithms—support vector regression (SVR), artificial neural network (ANN) with Elman network, artificial neural network with multilayer perceptron network, adaptive neuro-fuzzy inference system (ANFIS), and local linear regression, which are useful in river basins with scarce hydrological data. Daily, weekly, and monthly runoff and sediment yield (SY) time series of Vamsadhara river basin, India for a period from 1 June to 31 October for the years 1984 to 1995 were simulated using models based on these multiple machine learning algorithms. Simulated results were tested and compared by means of three evaluation criteria, namely Pearson correlation coefficient, Nash–Sutcliffe efficiency, and the difference of slope. The results suggested that daily and weekly predictions of runoff based on all the models can be successfully employed together with precipitation observations to predict future sediment yield in the study basin. The models prepared in the present study can be helpful in providing essential insight to the erosion–deposition dynamics of the river basin.

**Keywords:** tropical Indian river; simulating runoff and sediment yield; SVM; MLP; Elman; ANFIS; LLR

## 1. Introduction

The erosion–deposition dynamics in a river basin at both spatial and temporal scales play a vital role in shaping the hydrological processes occurred in the river basin [1–3]. Tropical Indian rivers are characterized by high and very low discharge during the summer and other months, respectively [4]. These rivers are also known for carrying sediment load to the oceans [5–8]. Reservoirs and channels associated with tropical rivers experience a

degraded surface water quality. Therefore, runoff and sediment yield simulation in short-term and long-term periods in tropical rivers is of crucial importance in water resource management projects. The complex, nonlinear, and variability nature of the sediment yield (SY) and runoff processes depend on precipitation patterns and river basin characteristics. There are a number of other elements that make it difficult to forecast with great precision. In recent decades, different types of models have been developed by hydrologic modelers and professionals. The physically based models simulate the runoff and SY, if enough data exist. The lack of sufficient input data restricts the using of these models in low-monitoring areas. Hence, a number of models have been developed, which enable us to simulate SY and runoff with the existing dataset. Artificial neural networks and soft computing techniques have proven to be decent alternatives and have been employed by researchers for modeling rainfall–runoff–sediment yield processes [1,3,5,9–28]. Data-scarce river basins can take advantage of these popular soft-computing techniques for modeling purposes.

Machine learning models have the ability to effectively capture chaotic and hidden time-series patterns. Precipitation dynamics combined with landscape attributes control the runoff and sediment yield. Tropical Indian rivers are known to receive their precipitation from monsoon winds. Tropical cyclones also bring highly intensive flooding rainfalls. These basins cause high runoff irrespective of basin area [29,30]. These high intensity rainfall events result in high runoff and soil erosion. Sediment yield is, thus, very much related to the rainfall pattern and intensity [31]. In the present study, a tropical eastern Indian river basin, Vamsadhara river basin, was chosen as a case study and the rainfall–runoff–sediment yield dynamics were modeled based on the existing dataset. A comparative analysis among five ML algorithms, including SVR, ANN with Elman network (Elman), ANN with multilayer perceptron network (MLP), ANFIS and local linear regression (LLR), was made to identify the efficiency of these methods in capturing the complex runoff–SY process. The primary objective of this article is to evaluate performances of SVM, ANNs (MLP and Elman), ANFIS, and LLR techniques and compare them in terms of statistical criteria, emulating runoff and sediment yield at daily, weekly, and monthly timescales. Ref. [23]'s dataset was applied to a basin runoff and SY simulation to compare the performance of SVM with other machine learning algorithms as stated before. The tropical eastern Indian river basin is an important river for prospective river-linking projects in India. Devising a model based on historical data can help policymakers and decisionmakers to have a good understanding of infrastructure development in the basin area from an erosion–deposition perspective.

## 2. Materials and Methods

### 2.1. Description of Methods

The relationships among rainfall, runoff, and sediment yield at different temporal scales were analyzed using multiple machine learning algorithms. The methods employed in this study are as follows:

(1)  Support Vector Regression (SVR): Support vector machines (SVMs) have been well described by [32–36]. Learning in SVMs is very robust from the point of view of the precision of the computations [37,38]. Ref. [23] stated that, in SVR, employing a nonlinear mapping function (kernel function) results in having to work in a higher dimensional feature space, consequently overcoming the dimensionality issue. [39,40] proved that the Gaussian radial basis kernel function (GRKF) is superior and more common than kernel function.

(2)  Artificial Neural Networks (ANNs): A learning process makes neural networks able to conduct information processing. This process is a link to weights adaptation, which causes producing an approximate output(s) by the network. Indeed, an ANN learning process will improve the system's correct response to an input via the development of a strengthened current matrix of nodal weights [41]. We used two different networks for this study:

MLP Networks: They are the most common and popular of all ANNs. MLP networks have 3 layers or more: an input layer, which is used to present data to the network; an output layer, which is used to produce an appropriate response(s) to the given input; and one or more intermediate layers, which are used to act as a collection of feature detectors (Figure 1). MLP networks have been successfully used earlier in hydrological studies in different river basins across the world [11,42–48].

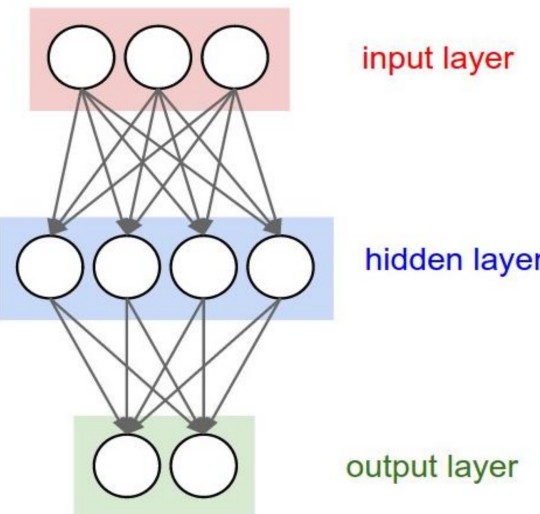

**Figure 1.** Illustration of an MLP network.

Elman Networks: They are a type of recurrent neural network with connections back to a specific copy layer from its hidden layer (Figure 2). This means that the network's learned function can be based on the current inputs as well as a history of the network's past state(s) and outputs. In other words, an Elman network is a finite state machine that learns which states to remember (i.e., what is relevant). Because the particular copy layer is considered as just another set of inputs, normal back-propagation learning techniques may be utilized (something that recurrent networks cannot perform). A duplicate of the concealed layer units is created to a copy layer at each time step [49]. Elman networks are an extension of two-layer sigmoid linear architecture [50,51]. Hence, they inherit the capability to fit any input/output function with a finite number of discontinuities. They can also fit temporal patterns, although a complicated function may need many neurons in the recurrent layer. In addition, the recurrent model takes significantly longer to train than the MLP model because of its more complicated design [52]. The use of Elman network in hydrological studies is, however, very limited [53,54].

(3) Adaptive Neuro-Fuzzy Inference System (ANFIS): [55] proposed ANFIS, a universal approximator capable for approximating any actual continuous function on a compact set to any degree of precision. ANFIS is made up of nodes that have defined tasks or obligations, which are organized into layers with specified functions. To select a set of parameters, it employs a hybrid learning method that combines back-propagation gradient descent error digestion and the least-squares technique. It may be used to build a collection of fuzzy "If–Then" rules with suitable membership functions in order to produce the input–output pairings that have been pre-specified [56–58]. Earlier, ANFIS has been used in several hydrological studies successfully [59–63]. The Gaussian membership function is adopted for the present study. For details of theoretical background, interested readers can refer to [55].

(4) Local Linear Regression (LLR): Like neural network models, LLR is a non-parametric method that does not require to be trained. Many difficulties with low-dimensional forecasting and smoothing have been solved using the LLR method. To create an initial forecast, the LLR technique simply requires three data points, after which it uses all newly updated data as it is able to make future forecasts [64]. LLR uses a set

of $P_{max}$ closest points to a query point to perform linear regression, which results in the creation of a linear model in the query point's neighborhood [65]. The kd-tree (k-dimensional tree) technique, a rapid closest neighbor search algorithm detailed in [66], is used to calculate the $P_{max}$ nearest points to the query point. The linear matrix problem is solved when given a neighborhood of $P_{max}$ points:

$$Xm = y \tag{1}$$

where $X = p_{max} \times d$ matrix of the $P_{max}$, $y$ = column vector of length $P_{max}$, and $m$ = column vector of parameters that must be determined to provide the optimal mapping from $X$ to $Y$, such that:

$$
\begin{pmatrix}
x_{11} & x_{12} & x_{13} & \cdots & x_{1d} \\
x_{21} & x_{22} & x_{23} & \cdots & x_{2d} \\
\vdots & \vdots & \vdots & \ddots & \vdots \\
x_{x_{pmax}1} & x_{x_{pmax}2} & x_{x_{pmax}3} & \cdots & x_{x_{pmax}d}
\end{pmatrix}
\begin{pmatrix}
m_1 \\
m_2 \\
m_3 \\
\vdots \\
m_d
\end{pmatrix}
=
\begin{pmatrix}
y_1 \\
y_1 \\
\vdots \\
y_{pmax}
\end{pmatrix} \tag{2}
$$

**Figure 2.** Illustration of an Elman network.

The number of linearly independent rows in the matrix $X$ determines whether or not the solution for m exists or is unique [64,65].

If the matrix $X$ is square and non-singular, then the unique solution to Equation (1) is $m = X^{-1}y$. If $X$ is not square or singular, we modify Equation (1) and attempt to find a vector $m$, which minimizes:

$$|Xm - y|^2 \tag{3}$$

Ref. [67] demonstrated that a pseudo-inverse matrix is the only answer to this problem [64,65,67]. The LLR technique has the benefit of enabling us to carry out statistical modeling locally with a small quantity of sample data. In locations of high data density in the input space, LLR can provide extremely accurate predictions. Ref. [66] can be consulted for more information on LLR.

## 2.2. Model Performance Indices

Three performance indices—Pearson's correlation coefficient (*r*), Nash–Sutcliffe efficiency (E), and the difference of slope (S$_{Diff}$)—were used in the present study for evaluating the performance of the models. Nash–Sutcliffe efficiency indicator has been frequently used to assess the effectiveness of hydrologic models. E value can vary between $-\infty$ and 1, where an efficiency of 1 indicates that the observed and estimated values are in complete agreement. E = 0 shows that all estimated values are equal to the observed values' mean. The mean of the observed values is a better predictor than the predicted values when E is negative. The efficiency coefficient outperforms correlation-based sensitivity requirements for observed and calculated means and variances, but is too sensitive to outliers [68,69]. The idea behind S$_{Diff}$ is that, although r indicates a model's variational accountability and E represents its efficiency, when observed and projected values are compared, there is no corresponding metric for the degree of predictability of a best-fit model to the 1:1 line. As a consequence, S$_{Diff}$ was defined as the distance between the slope of a best-fit line on a scatter plot of expected vs. observed data for a particular model and the 1:1 line on a scatter plot of predicted vs. observed data. An S$_{Diff}$ of 0% means that the scatter plot's best-fit line is parallel to the 1:1 line, guaranteeing that the best-fit linear model is fully predictable. If S$_{Diff}$ is 100 percent, the best-fit line is the average line with a zero slope. The scatter plot's best-fit linear model would exaggerate the low observed values while underestimating the high ones if S$_{Diff}$ was between 0% and 100%. According to a negative S$_{Diff}$ score, the best-fit linear model of the scatter plot would underestimate the low observed values and overestimate the high ones. The mathematical representation of the performance indices are as follows:

$$r = \frac{\sum_{i=1}^{n} \left[ (Q_{pi} - Q_{pavg})(Q_{oi} - Q_{oavg}) \right]}{\sqrt{\left[ \left( \sum_{i=1}^{n} (Q_{pi} - Q_{pavg})^2 \right) \left( \sum_{i=1}^{n} (Q_{oi} - Q_{oavg})^2 \right) \right]}} \tag{4}$$

$$E = 1 - \frac{\sum_{i=1}^{n} (Q_{pi} - Q_{oi})^2}{\sum_{i=1}^{n} (Q_{oi} - Q_{oavg})} \tag{5}$$

where, $Q_{oi}$ and $Q_{pi}$ are the observed and predicted (or simulated) values, respectively. $Q_{oavg}$ and $Q_{pavg}$ are the mean values of the observed and predicted values, respectively.

## 2.3. Study Area and Data Used

The present study was conducted in the Vamsadhara river basin, which is located between the Mahanadi and Godavari river basins in eastern India (Figure 3). The region lies between $18°15'$ and $19°55'$ north latitudes, and $83°20'$ and $84°20'$ east longitudes. The south–west monsoon and the rare cyclones generated owing to the depression in the Bay of Bengal impact the precipitation in the basin from June to October.

There are two types of climates in this river basin. The coastal area has a semi-arid climate, while the upper reaches have a dry sub-humid climate. The Vamsadhara river rises near Lanjigarh in Odisha and flows for 254 km before joining the BoB at Kalingapatnam in Andhra Pradesh. It has a basin area of 10,450 square kilometers. The average rainfall amount in the basin is 940.2 mm near the coast, 1551.6 mm in the northeast, and 1250.2 mm in the northwest [70]. For the Vamsadhara river basin, the annual average maximum temperature is about 32 °C and minimum temperature is about 21 °C. The elevation ranges in the Vamsadhara river basin from 10 m above MSL in the south near the coast to 1545 m in the northwest (hills near Bissam Cuttack). The reservoirs of Badnalla and Harabhangi are located within the Kashinagar gauge station, while the Gotta barrage is located outside the gauge station. The annual river flow and sediment load are about 5943.56 m$^3$/s and 7511.54 kg/s, respectively. Clayey loam soils cover the majority of the Vamsadhara river basin. Over this basin, major land is occupied by forests (52%) and agricultural lands (30%).

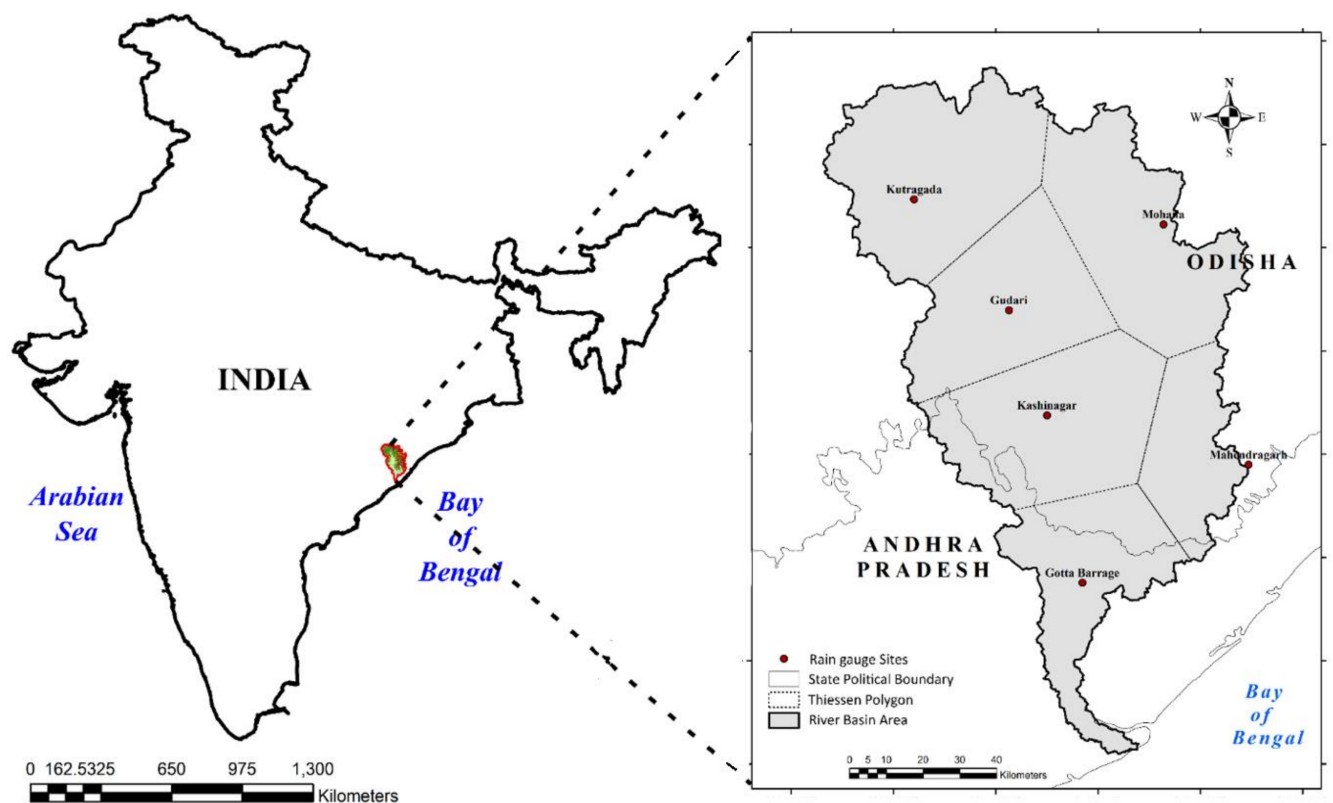

**Figure 3.** Location of Vamsadhara river basin in India and location of rain gauge sites with Thiessen polygons within the river basin.

The study basin includes six rain gauge stations (Kutragada, Gudari, Kashinagar, Gotta Barrage, Mohana, and Mahandragarh) and the weighted rainfall for the study region was estimated using the Thiessen polygon method, as shown in Figure 3. Daily rainfall, runoff, and SY data were obtained for a period from 1 June to 31 October for the years 1984 to 1995. The precipitation from monsoon winds or cyclonic depressions is concentrated during these months in the study basin. The rainfall, runoff, and SY time series of the monsoon period (June–October) for the years of 1984–1987 was employed to train the SVM, ANN, ANFIS, and LLR models, and the time series of 1988–1989 and 1992–1995 were considered for their testing and validation, respectively. The input variables with different time lags selected for the SVM, ANN, ANFIS, and LLR models based on the correlation coefficient method for predicting runoff and SY (daily, weekly, and monthly) are given in Table 1.

**Table 1.** Input variables used for SVM, ANN, ANFIS, LLR, and MRPRT models (Ri = total rainfall in mm; Qu = runoff in $m^3/s$; Sy = sediment yield in kg/s; t = current day; I/P = input; and O/P = output).

| Time Scale | Variable | Input − Output Variables with Lag Times | | | | | |
| --- | --- | --- | --- | --- | --- | --- | --- |
| | | Runoff | | | | Sediment Yield | |
| | | t | t − 1 | t − 2 | t − 3 | t | t − 1 |
| Daily | Ri | I/P | I/P | - | - | I/P | I/P |
| | Qu | O/P | I/P | I/P | I/P | I/P | - |
| | Sy | - | - | - | - | O/P | I/P |
| Weekly | Ri | I/P | I/P | - | - | I/P | - |
| | Qu | O/P | I/P | - | - | I/P | - |
| | Sy | - | - | - | - | O/P | - |

**Table 1.** *Cont.*

| Time Scale | Variable | Input − Output Variables with Lag Times | | | | | |
| | | Runoff | | | | Sediment Yield | |
| | | t | t − 1 | t − 2 | t − 3 | t | t − 1 |
| Monthly | Ri | I/P | I/P | I/P | - | I/P | - |
| | Qu | O/P | - | - | - | I/P | - |
| | Sy | - | - | - | - | O/P | - |

## 3. Results and Discussion

All the datasets were normalized between 0 and 1 prior to feeding to the models. According to Table 1, daily values of total rainfall with 0 to 1 lag days (Ri(t) and Ri(t − 1)), daily values of runoff with 1 to 3 lag times (Qu(t − 1), Qu(t − 2), Qu(t − 3)), daily values of sediment yield with 0 to 1 lag time (Sy(t) and Sy(t − 1)), weekly values of total rainfall with 0 to 1 lag week (Ri(t) and Ri(t − 1)), weekly values of runoff with 1 lag week (Qu(t − 1)), current week sediment values (Sy(t)), monthly values of total rainfall with 0 to 2 lag months (Ri(t), Ri(t − 1), Ri(t − 2)), and monthly values of runoff (Qu(t)) and sediment (Sy(t)) on the current month were used as the inputs of the models for predicting current runoff and sediment (as the output variables).

Four years of data (1984–1987) were used to train the models and two years of data (1988–1989) were utilized to test the models' performance. The data over the next four years (1992–1995) were forecasted by the models. The performance metrics (r, E, and $S_{Diff}$) were used to compare the observed and estimated data from 1992 to 1995. All the predictions and calculations were conducted in MATLAB 2016a software.

Runoff prediction: The validation and testing results obtained for runoff at the three different time scales—daily, weekly, and monthly—for all the models are given in Table 2.

**Table 2.** Comparison of the performance of SVM, ANN, ANFIS, and LLR models for daily, weekly, and monthly runoff prediction in testing (1988–1989) and validation (1992–1995) periods.

| Runoff Models | | | Testing (1988–1989) | | Validation (1992–1995) * | | |
| | | | r (%) | E (%) | r (%) | E (%) | Sdiff (%) |
| Daily | ANNs | SVM | 86.1 | 72.6 | 92.10 | 80.02 | 33.79 |
| | | MLP | 86.3 | 73.3 | 90.1 | 72.7 | 40.15 |
| | | Elman | 87.66 | 76.13 | 92.99 | 83.94 | 25.53 |
| | ANFIS | | 89.64 | 79.99 | **94.62** | 88.78 | 14.35 |
| | LLR | | 83.87 | 68.80 | 94.40 | **89.01** | **9.12** |
| Weekly | ANNs | SVM | 80.67 | 62.03 | 91.16 | 67.55 | 47.0 |
| | | MLP | 79.6 | 60.3 | 87.4 | 54.6 | 56.95 |
| | | Elman | 77.57 | 39.79 | 90.35 | 79.84 | 27.15 |
| | ANFIS | | 87.10 | 75.55 | 90.99 | 77.05 | 28.96 |
| | LLR | | 71.39 | 47.51 | **94.44** | **88.67** | **16.79** |
| Monthly | ANNs | SVM | 81.47 | 24.31 | 86.33 | 14.13 | 60.09 |
| | | MLP | 77.4 | 26.0 | 79.3 | −4.2 | 74.98 |
| | | Elman | 73.82 | 41.05 | 75.53 | 45.98 | 42.61 |
| | ANFIS | | 88.75 | 78.93 | 80.00 | 39.24 | **11.6** |
| | LLR | | 90.98 | 67.74 | **86.62** | **69.18** | 33.77 |

* The best performance index for a particular model for the validation data has been shown in bold.

At daily scale, the LLR model outperforms the other models based on the E (89.01%) and sdiff (9.12%) criteria. However, the ANFIS model shows a better performance than the LLR according to the r value (94.62%). Among the intelligent models, the ANFIS model performs successfully based on all the criteria. The Elman model follows the ANFIS as a second best intelligent model.

At weekly scale, the LLR model outperforms the other models based on the r (94.44%), E (88.67%), and sdiff (16.79%) criteria. Among the intelligent models, the Elman model performs accurately based on the E (79.84%) and sdiff (27.15%) criteria. The ANFIS model follows the Elman as a second best intelligent model.

At monthly scale, the LLR model outperforms the other models based on the r (86.62%) and E (69.18%) criteria. However, the ANFIS model shows a better performance than the LLR according to the sdiff value (11.6%). Among the intelligent models, the ANFIS model performs successfully based on the r (80%) and sdiff criteria. The Elman model follows the ANFIS as a second best intelligent model.

In general, the ANFIS and Elman techniques showed high ability in daily, weekly, and monthly runoff estimating. However, the non-intelligent LLR model showed notable accuracy in its estimations at different time scales.

From daily to monthly scales, the MLP model revealed a 10.8% reduction in performance based on the r criterion. A similar trend can be seen for the E performance metric of 76.9% for the MLP. The Elman model provided a reduced performance of 17.46% from daily to monthly predictions based on the r measure. A similar decreasing trend of 37.96% is observed according to the E performance measure for the Elman neural network model. ANFIS model showed a reducing performance of 14.62% and 49.54% in the r and E values, respectively, from daily to monthly scales. The LLR model performance (r criterion) for weekly runoff predictions was slightly better than daily runoff values, but the LLR provided a reduced performance of 7.82% between weekly and monthly predictions. A decreasing trend is observed from daily through monthly predictions based on the reduced E measure (about 19.83%).

In general, all of the applied intelligent and non-intelligent models showed a decreasing performance for runoff predictions from daily to monthly scales.

Figures 4–6 show the scatter plots of the models for runoff prediction at daily, weekly, and monthly scales, respectively. According to these figures, the LLR model is closer to the line 1:1 than the other models at daily and weekly scales. At monthly scale (Figure 6), the ANFIS model is close to line 1:1. The ANFIS and Elman models are closer to line 1:1 than the other intelligent models.

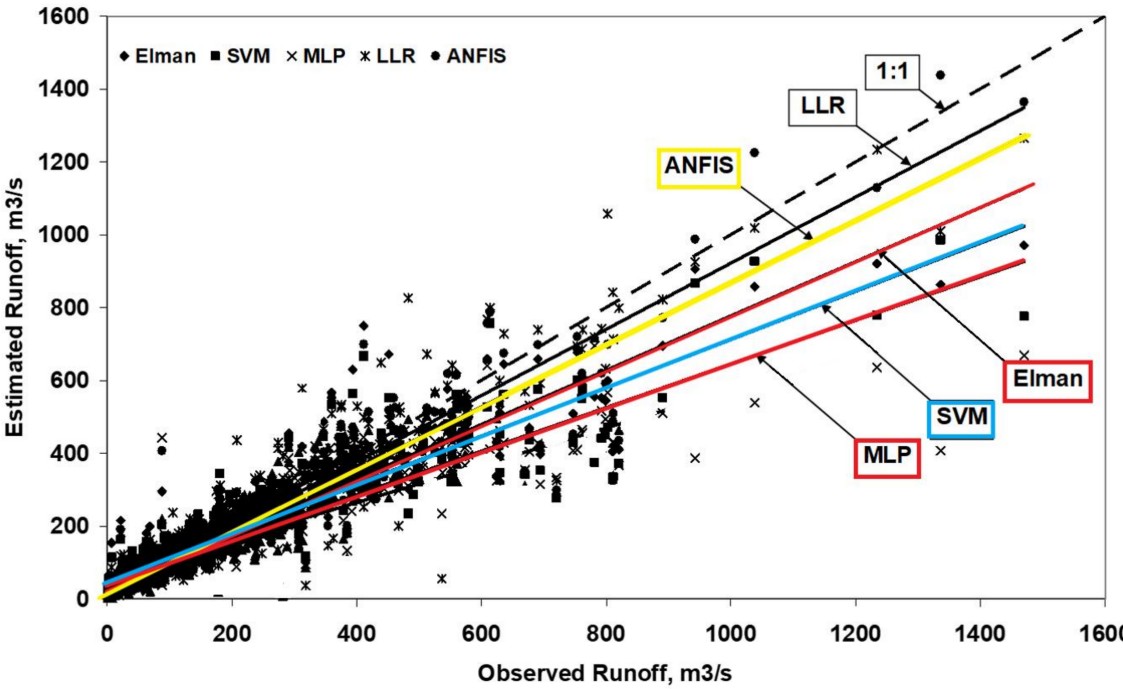

**Figure 4.** Scatter plot of the observed vs. estimated daily runoff (1992–1995) using SVM, ANN, ANFIS, and LLR methods.

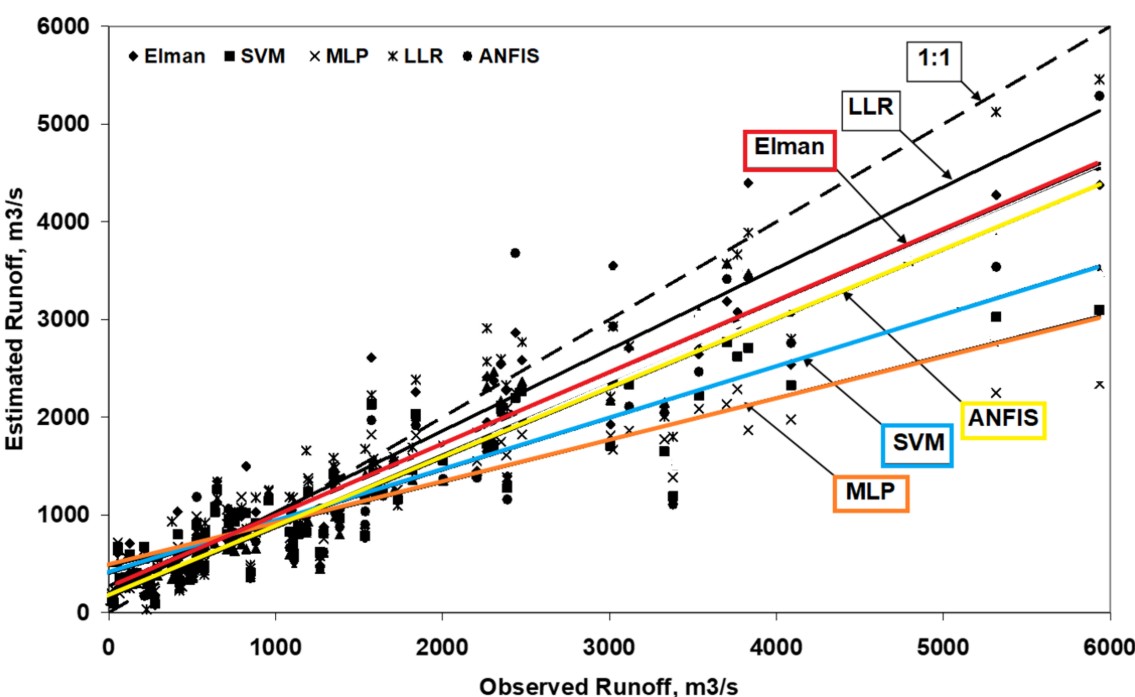

**Figure 5.** Scatter plot of the observed vs. estimated weekly runoff (1992–1995) using SVM, ANN, ANFIS, and LLR methods.

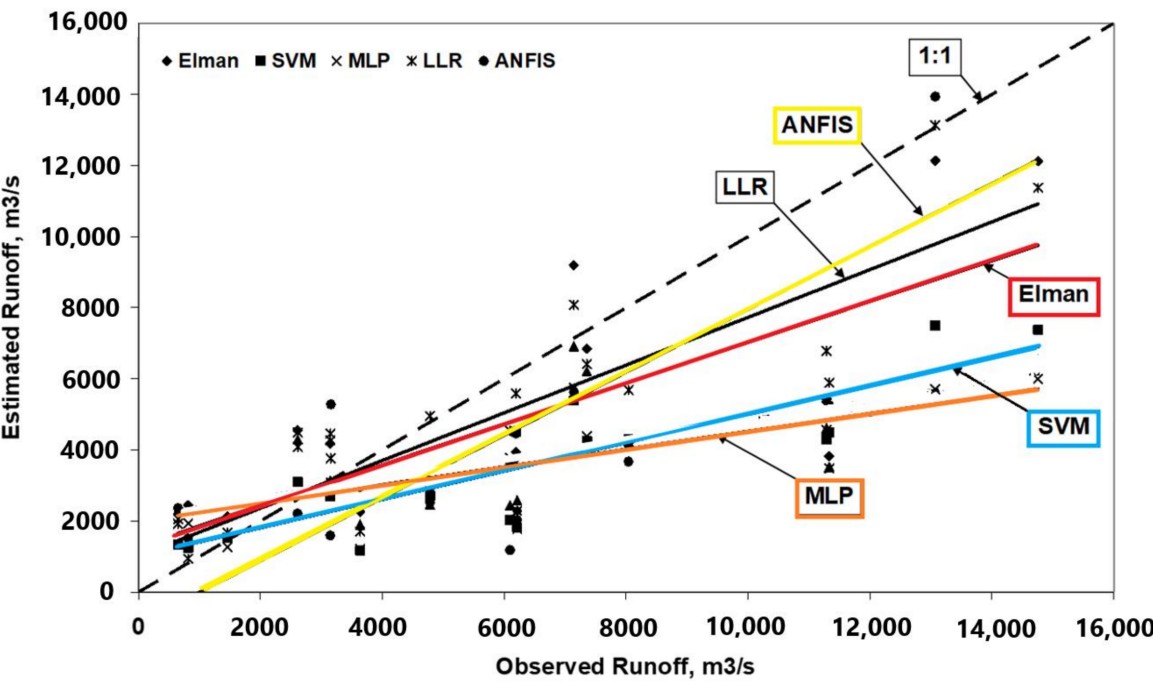

**Figure 6.** Scatter plot of the observed vs. estimated monthly runoff (1992–1995) using SVM, ANN, ANFIS, and LLR methods.

Sediment yield prediction: The same models were applied in order to predict daily, weekly, and monthly sediment yields in the study basin. Testing and validation results are presented in Table 3.

**Table 3.** Comparison of the performance of SVM, ANN, ANFIS, and LLR models for daily, weekly, and monthly sediment yield prediction in testing (1988–1989) and validation (1992–1995) periods.

| Sediment Yield Models | | | Testing (1988–1989) | | Validation (1992–1995) * | | |
|---|---|---|---|---|---|---|---|
| | | | r (%) | E (%) | r (%) | E (%) | Sdiff (%) |
| Daily | | SVM | 80.02 | 62.94 | 87.87 | **75.68** | 30.46 |
| | ANNs | MLP | 79.30 | 62.80 | 83.20 | 68.00 | 21.18 |
| | | Elman | 77.63 | 53.01 | **87.99** | 36.30 | −29.61 |
| | ANFIS | | 88.10 | 76.87 | 87.17 | 73.09 | **12.11** |
| | LLR | | 93.20 | 86.25 | 82.18 | 66.78 | 27.03 |
| Weekly | | SVM | 78.43 | 60.26 | 88.08 | 74.62 | 34.84 |
| | ANNs | MLP | 80.20 | 64.10 | 75.10 | 51.80 | 45.90 |
| | | Elman | 79.75 | 57.98 | **90.03** | 43.69 | −21.97 |
| | ANFIS | | 83.12 | 67.19 | 83.13 | 67.55 | 79.15 |
| | LLR | | 82.05 | 66.10 | 89.88 | **78.99** | **12.43** |
| Monthly | | SVM | 81.32 | 62.44 | 87.66 | 74.49 | 24.76 |
| | ANNs | MLP | 89.40 | 79.10 | 74.10 | 53.70 | 52.06 |
| | | Elman | 89.13 | 79.31 | **88.99** | **76.25** | **9.01** |
| | ANFIS | | 99.94 | 99.81 | 85.47 | −102.74 | −62.28 |
| | LLR | | 92.10 | 70.19 | 52.77 | 16.84 | 75.61 |

\* The best performance index for a particular model for the validation data are emboldened.

At validation stage and daily scale, the Elman model with an r value of 87.99%, SVM model with an E value of 75.68%, and the ANFIS model with an sdiff value of 12.11% outperform the other models. In general, the ANFIS model performs more precisely than the others based on all the criteria.

At weekly scale, the LLR model outperforms the other models based on the E (78.99%) and sdiff (12.43%) criteria values. However, the Elman model performs well based on the r measure (90.03%).

At monthly scale, the intelligent Elman model estimates sediment yield with a higher accuracy than the other models based on the r (88.99%), E (76.25%), and sdiff (9.01%) criteria. The SVM model follows the Elman as the second best model based on all the criteria.

From daily to monthly scales, the MLP model showed a 9.1% reduction in performance based on the r criteria. A similar pattern can be seen with the E and sdiff performance metrics for the MLP model. As the nonlinearity in the time series reduces, the MLP model's performance accuracy considerably reduces (from daily to monthly scales). The Elman model provided an increased r value of 1% from daily to monthly time scales. The E performance measure in the Elman neural network model shows a similar pattern with an increase of 39.95%. From daily to monthly scales, the ANFIS model showed a 1.7% reduction in performance (r criterion). A similar tendency can be seen for the E performance metric, while the E value of the ANFIS model for monthly predictions was considerably reduced, down to a negative value, which is not in accordance with Ref. [71]. The LLR model provided an increased r value of 7.7% between daily and weekly predictions, but a decreased value of 37.11% between weekly and monthly predictions. A similar trend with an increased E value of 12.21% between daily and weekly scales is observed for the LLR model, but the E values of the LLR model significantly decreased down to 62.15% from weekly to monthly scale.

In general, the performance accuracy of all the models in sediment yield prediction reduces when the time scale is increased.

Figures 7–9 illustrate the scatter plots of the applied models at the validation stage for sediment yield prediction at daily, weekly, and monthly scales, respectively. According to these models, the ANFIS, LLR, and Elman models are closer to line 1:1 than the other models at daily, weekly, and monthly scales, respectively. The MLP, Elman, and SVM

models are the second best models based on the short distance from line 1:1, at daily, weekly, and monthly scales, respectively.

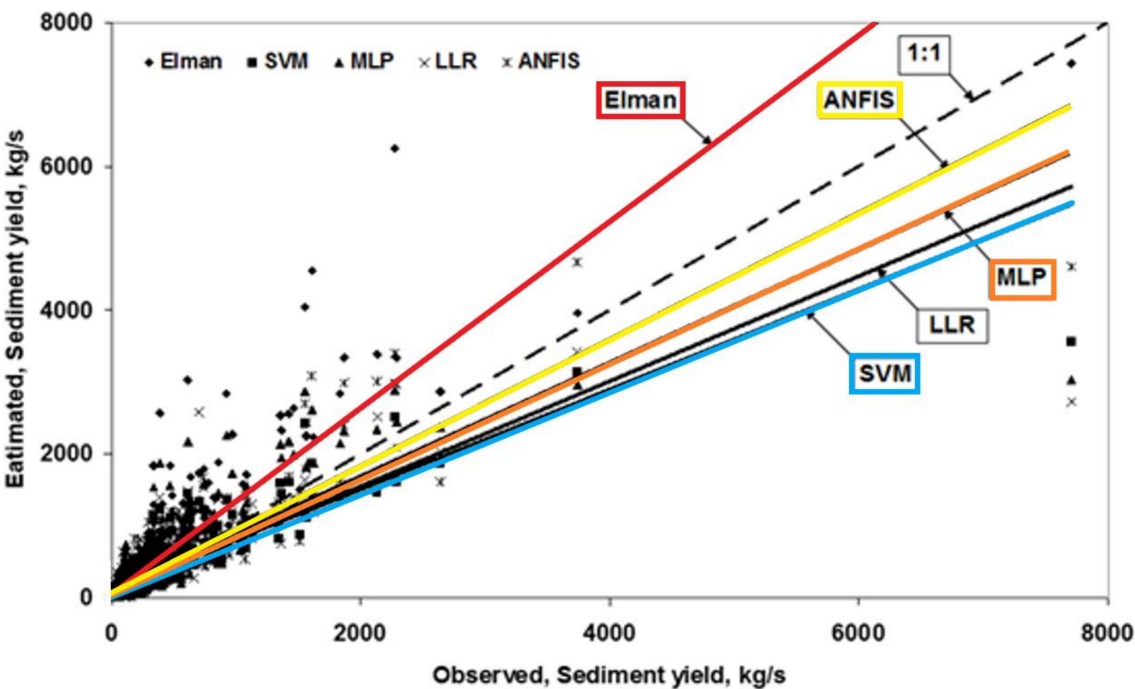

**Figure 7.** Scatter plot of the observed vs. estimated daily sediment yield (1992–1995) using SVM, ANN, ANFIS, and LLR methods.

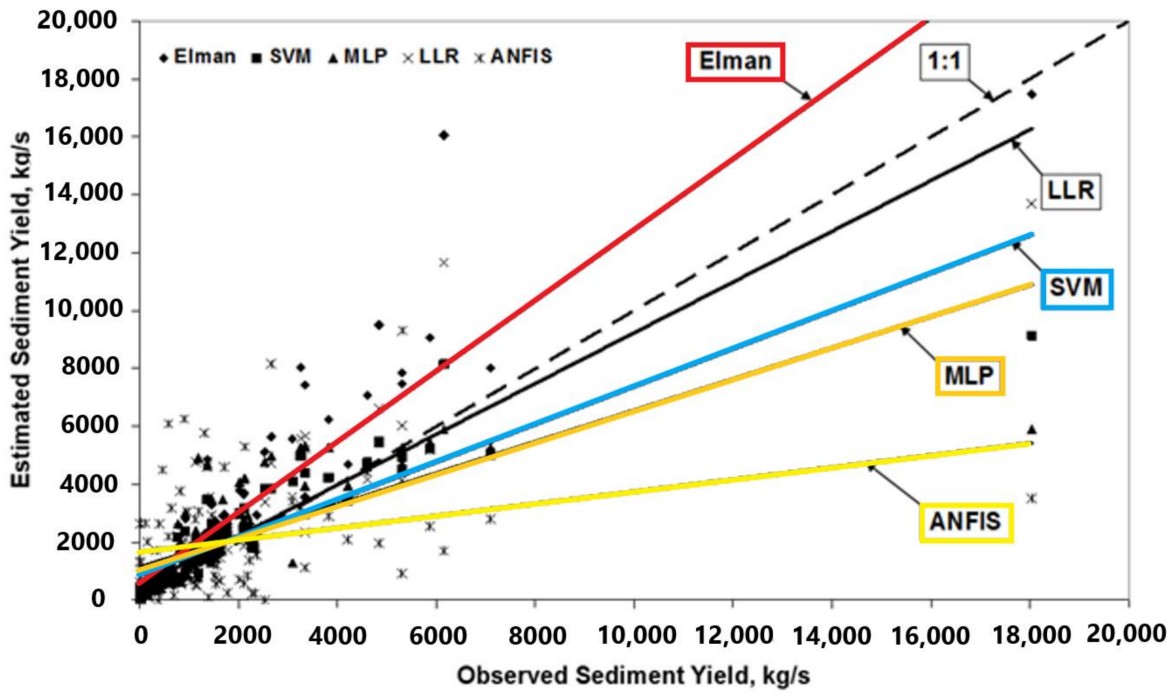

**Figure 8.** Scatter plot of the observed vs. estimated weekly sediment yield (1992–1995) using SVM, ANN, ANFIS, and LLR methods.

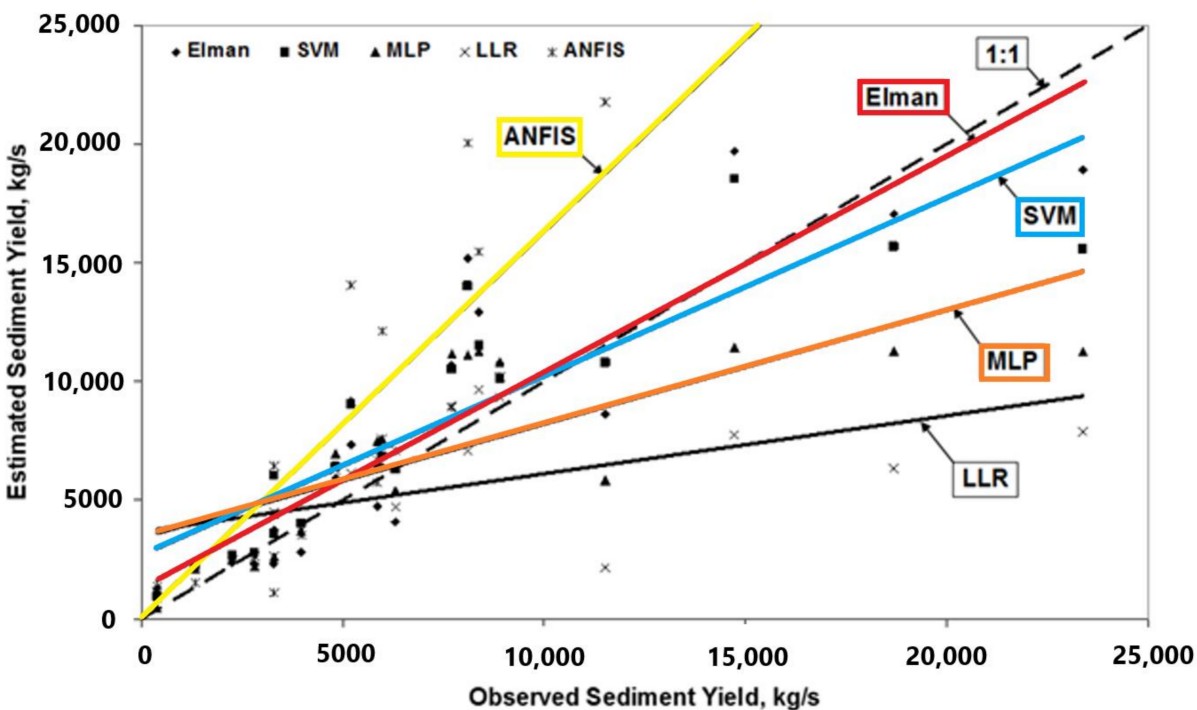

**Figure 9.** Scatter plot of the observed vs. estimated monthly sediment yield (1992–1995) using SVM, ANN, ANFIS, and LLR methods.

In general, it can be concluded that the intelligent models have high ability in sediment yield, estimating at different time scales.

For both runoff and sediment yield prediction, the intelligent Elman model showed less error in its estimations at different time scales. The SVM and MLP models were more suitable for sediment yield prediction than the runoff prediction. The ANFIS and LLR models were more suitable for runoff prediction than the sediment yield prediction. Since the intelligent models are more sensitive toward their input variables, input data length, and their training parameter values, considering unsuitable variables or values can affect the models' performance and increase errors more than the non-intelligent models would, such as LLR. The intelligent models were more successful than the LLR model in sediment yield prediction because of the high nonlinearity degree of the rainfall–runoff–sediment process. By increasing the time scales and reducing the nonlinearity degree of the processes and also input data length, the accuracy of all the applied models decreased. The findings of this study are in accordance with most of the previous research [46,72].

## 4. Conclusions

The present study attempted to establish the effectiveness of machine learning algorithms, SVM, MLP, Elman, and ANFIS, to capture variability in rainfall–runoff–sediment yield dynamics in a tropical Indian river basin. Runoff and sediment yield were estimated on three time scales, daily, weekly, and monthly, with different input combinations and lag times. The performance of the intelligent models were compared with the non-intelligent LLR model using three performance criteria: r, E, and sdiff. The results showed that the ANFIS and LLR models were more suitable for runoff prediction than the sediment yield prediction. The SVM and MLP models were more suitable for sediment yield prediction than the runoff prediction. For both runoff and sediment yield prediction, the intelligent Elman model showed high ability at different time scales. The intelligent models were more successful than the LLR model in sediment yield prediction. By increasing the time scales, the accuracy of all the applied models decreased. For increasing the models accuracy, it is recommended to use other hydrological data such as temperature, evaporation, and different lag times. Moreover, it is recommended to apply different optimization algorithms such

as genetic algorithm for the optimal determination of the intelligent models' parameters, improving their performance. Other intelligent models such as random forest are also recommended to apply and evaluate their efficiency in simulating the studied processes.

The models applied in the present study can be helpful in providing essential insight to the erosion–deposition dynamics of the Vamsadhara river basin. This study established the robustness of the multiple machine learning algorithms in capturing the variability of hydrological events in a tropical river basin.

**Author Contributions:** Conceptualization, D.M., A.M.N. and M.S.; methodology, D.M., A.M.N., M.H.K., M.G. (Mohsen Ghafari) and M.S.; software, D.M., A.M.N., M.H.K., M.G. (Mohsen Ghafari) and M.S.; data curation, M.H.K., M.G. (Mohsen Ghafari) and M.S.; writing—original draft preparation, D.M., A.M.N., M.H.K., M.G. (Mohsen Ghafari), M.S., M.S.J., M.T., S.T. and M.G. (Marzieh Ghodsi); writing—review and editing, D.M., A.M.N., M.H.K., M.G., M.S., M.S.J., M.T., S.T. and M.G.; visualization, D.M., A.M.N., M.H.K., M.G. (Mohsen Ghafari), M.S., M.S.J., M.T., S.T. and M.G. (Marzieh Ghodsi); project administration, D.M. and A.M.N. All authors have read and agreed to the published version of the manuscript.

**Funding:** This research received no external funding.

**Institutional Review Board Statement:** Not applicable.

**Informed Consent Statement:** Not applicable.

**Data Availability Statement:** Data will be made available on request.

**Conflicts of Interest:** The authors declare no conflict of interest.

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
