# Peer review of "Runoff and Sediment Yield Processes in a Tropical Eastern Indian River Basin: A Multiple Machine Learning Approach"

_land, doi:10.3390/land12081565_

Round 1

Reviewer 1 Report

This study attempts to model rainfall-runoff sediment yield with the help of five machine learning algorithms – Support Vector Regression, Artificial Neural Network with Elman network, Artificial Neural Network with multilayer perceptron network, Adaptive Neuro-fuzzy Inference System, and Local Linear Regression. The topic of this manuscript is overall interesting. Some modifications are recommended before publication.

1.     Line 33. “Runoff and sediment yield processes are vital for any river basin” is a general information and can be discarded.

2.     Lines 39-41. The means of the two sentences seem similar.

3.     Lines 60-62. Confusing. The description should closely focus on the topic of the study. Are “precipitation” and “infiltration” the topic of this study?

4.     Lines 63-64. The mean is unclear. “high and very low discharge”? “during the summer and other months”?

5.     The universal significance or implication beyond the local dimension of a Tropical Eastern Indian River Basin should be explained in the discussion and conclusion.

6.     The longitude and latitude in Figure 3 could be discarded.

 Overall, the Quality of English Language is ok.

Author Response

Please see response in attached document

Reviewer 2 Report

I thank the authors for using ML in sediment yield predictions. Given the significance of this topic, I have some observations: 

1. The abstract is poorly written without any substances. 

2. In the introduction section, you must add more on comparing the sediment yield from traditional hydrological models and ML based models. Also the connectivity between these two modeling aspects are missing. 

3. No description of data is presented in the manuscript. The landuse type, slope, DEM, and other data important for sediment yield from a watershed scale is not presented. I checked the paper from where the authors collected data, where they did not mention anything regarding the daily, weekly or monthly data. In your manuscript, the data source is also missing. 

5. How many data in training, testing and validation did you use? Nothing mentioned about it. In ML, if your data is small, you will not get desired results. So, you must include some section about that. Also the you just used rainfall data and predicting runoff. Is it the runoff from the basin or the flow data from the river? Given the location of the basin, it could be influenced by tidal forces. How do you handle that in your ML models? Though I'm repeating, write something about the process based modeling approach and ML approach (as you are saying it is a substitute) as I mentioned in the introduction section.  

6. You must include statistical analysis such as ANOVA/MANOVA or pairwise non-parametric analysis to describe the model performances rather than just saying this % increase or decrease, etc. 

7. Conclusion section requires improvement. It looks it is discussion rather than concluding remarks.   

Author Response

Please see response in the attached document

Reviewer 3 Report

Brief summary

This paper, entitled ‘Runoff and Sediment Yield Processes in a Tropical Eastern Indian River Basin: A Multiple Machine Learning Approach’ focuses modeling rainfall-runoff-sediment yield with the help of five machine learning (ML) algorithms (Support Vector Regression (SVR), Artificial Neural Network (ANN) with Elman network, Artificial Neural Network with multilayer perceptron network, Adaptive Neuro-fuzzy Inference System (ANFIS), and Local Linear Regression) in the Vamsadhara river basin located in the eastern tropical region of India.

 Three performance indicators were used for evaluating the accuracy of the applied models:  Pearson’s correlation coefficient (r), Nash-Sutcliffe Efficiency (E), and difference of slope (SDiff).

They used daily observations of rainfall from six rain gauges and the runoff and sediment yield for the period 1984-1995. The data was used for training, testing and validating of the models.

 The authors concluded that daily and weekly predictions of runoff based on all the models can be successfully employed together with precipitation observations to predict future sediment yield in the study basin. Therefore, the models prepared in the present study can be helpful in providing essential insight to the erosion-deposition dynamics particularly in river basins with scarce observed data.

The list of literature is balanced and updated.

Main comments

Abstract:

-       - Reduce the text between the lines 33 and 41

-        - Add infos on the used data, evaluation parameters, etc.

-        Stress on the usefulness of the models in river basins with scarce observation data

Introduction:

 Please concentrate on the state of the art / literature review related to the addressed topic. You devoted almost half of the Introduction to introduce the study area and the used models which you may move to the dedicated sections.

 Materials and methods:

 The models and the performance indicators were well described. However, you provided very limited information on the study river basin.  Please provide more details on:

-          Climate /geography, hydrology / hydrological network, etc.

-          Land use / vegetation cover

-          Hydraulic structures (dams, etc.)

-          Population and main economic activities

-          Observed data: runoff gages (number, locations, instruments), sediment gages (number, locations, instruments), who collected the data, how data was published and shared, etc.

 Results and discussion

 This section needs to be completely revised:

-       - Most of this section is to move to methodology

-        - You need to concertante here on presenting the results and making comparison with the similar studies

 Conclusion

 Here also you mix conclusion with discussion. Revisit the text and at end provide some future prospects for this research work.

 Questions

-        - Why you used only the data from June to October ?

-        Why the observed data stopped in 1995 and we are in 2023 ?

-        - The models have been trained for specific conditions, what would be the impacts of the changes in : land use/land cover, hydraulic structures (new dams, etc.), climate change, etc. ?

 Minor comments

-        - Use colors in the graphs

-        - As done with SY, add graphs on runoff

Author Response

(The authors gave the same response as above.)

Reviewer 4 Report

The study attempts to model rainfall-runoff-sediment yield with the help of five traditional machine learning (ML) algorithms. In this manuscript the authors do not have any improvement of the algorithm model, and there is less innovation. Meanwhile, there are many similar studies in the published papers. Moreover, the figures in the manuscript are very rough. In my opinion, the manuscript should not be published in the journal.

Language is ok

Author Response

(The authors gave the same response as above.)

Round 2

Reviewer 2 Report

I still see some of my comments have not been addressed properly. However, the authors are claiming, those have been incorporated. 

Reviewer 3 Report

I think the authors addressed most of my comments.

Reviewer 4 Report

The manuscript can be published on the journal of Land now.

  •  
  •